# ARE LARGE LANGUAGE MODELS REALLY RELIABLE ZERO-SHOT TIME SERIES FORECASTER? FAILURE ANALYSIS VIA THE LENS OF STATIONARITY

## ABSTRACT

Large Language Models (LLMs) have been widely adapted as zero-shot time series forecasters recently. However, the reliability of such approach is still under debate. To address this, we propose a novel, simple and rigorous evaluation methodology on zero-shot LLM forecasters via the lens of stationarity. An unbiased and robust zero-shot forecaster should preserve stationarity, i.e. given an input series with distinct time-independent mean and variance, a capable forecaster should maintain these same data characteristics in the output. Our comprehensive experiments discover that LLMs consistently fail on preserving stationarity, with forecasts contaminated by profound hidden biases and trends persistently visible even after averaging over hundreds of iterations. Furthermore, the reasoning content of LLMs reveals that LLMs incline to blind guessing simplistic numeric patterns through the last few time steps of the input series, without genuine understanding of the full time series. Our findings underscore the crucial need for cautious application of LLMs in zero-shot time series forecasting. Our code repository will be publicly available upon publication of the paper.

## 1 INTRODUCTION

Time series forecasting drives critical applications across science, engineering, and business, spanning retail sales analysis (Böse et al., 2017), financial modeling (Sezer et al., 2020), traffic forecasting (Kablaoui et al., 2024), and healthcare monitoring (Morid et al., 2023a). The advent of Large Language Models (LLMs) like GPT (Achiam et al., 2023), Llama (Touvron et al., 2023), and DeepSeek (Liu et al., 2024a), renowned for their exceptional reasoning and generalization capabilities, has sparked their application in time series tasks like classifying, forecasting, and detecting anomalies (Chang et al., 2023; Zhou et al., 2023; Cao et al., 2024; Jin et al., 2024a;b; Liu et al., 2024c; Sun et al., 2024). A standout approach, **zero-shot** forecasting, which harnesses pretrained LLMs' language-based generalization capabilities to model time series without requiring any further reprogramming and finetuning, is gaining traction in academia and industry through methods like LLMTime (Gruver et al., 2023) and PromptCast (Xue & Salim, 2023).

Although these studies suggest that LLMs, excelling at handling sequential dependencies in text, might extend this capability to time series dependencies, direct validations between language modeling and time series forecasting remain largely underexplored. Enthusiasm for LLM-based forecasting contrasts with emerging skepticism. For example, in the **few-shot** (a.k.a. reprogrammed and finetuned) LLM forecasting approach, Tan et al. (2024) performed detailed ablations and discovered that simple patched attention encoders rival few-shot LLM forecasters with far less computation, challenging their necessity. However, the validity of the zero-shot approach remains unresolved.

**To answer this debate, we propose a novel, simple and rigorous approach to examine the validity of zero-shot LLM time series forecasters via the lens of stationarity.** Stationarity is a fundamental property in time series analysis, whose strong form is too restrictive and seldom employed in the real world, but its weak form is widely applied (Petrică et al., 2017), for example, in financial and economic modeling (Mills & Markellos, 2008), medical science (Morid et al., 2023b) and environmental detections (Baek et al., 2009). For brevity, stationarity refers to weak stationarity from now on. In layman's terms, stationary means that the mean, variance and autocovariance

of a time series are time-invariant. Such property is highly desired in time series analysis because it entails that the future can be predicted from the past (Dare et al., 2022). Therefore, although most real-world time series are non-stationary, they are often transformed by iterative differencing or Seasonal-Trend decomposition using Loess (STL), to stationary series for further analysis (Whittle, 1953; Dickey & Pantula, 1987; Hamilton, 2020). Traditional time series methods, like ARIMA (Autoregressive Integrated Moving Average), are built upon stationarity (Zhang, 2018).

**We propose to judge whether LLMs are reliable zero-shot forecasters by evaluating if their forecasts preserve stationarity when prompted with distinctly stationary inputs.** If the input time series is stationary and contains a distinct mean and variance, an unbiased and robust zero-shot forecaster should generate forecasts with the same mean and variance as the input, without any hidden trends or biases contaminating the forecasts. For example, the left hand side of Figure 1 shows an input series moving around 0 with distinct mean and variance, for

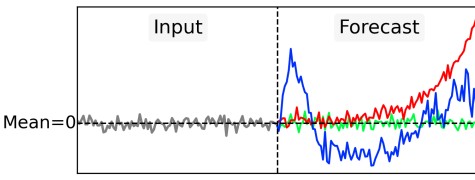

Figure 1: Motivation of stationarity preservation evaluation: Sensible (green) vs. Unreliable (red and blue) forecasts.

which it is natural for us to expect the forecasts on the right hand side to be unbiased with similar mean and variance as in green color (which is basically the core idea of stationarity). If the forecasts, however, appear as in red or blue colors, we would undoubtedly discard them and deem the forecasters unreliable.

Some may argue that accuracy is a more common and standard metric so why aren't we using accuracy for evaluation? Because **LLMs are the first ever textual models boasting zero-shot capabilities across diverse tasks, including time series forecasting**. Before LLMs, no textual models have ever claimed extraordinary generalization capabilities to perform well on time series tasks without specialized training. Traditional models are trained and tested with specific datasets (Lim & Zohren, 2021; Benidis et al., 2022). For example, if a model is trained with train-set from source A, then we will use the corresponding test-set from A for evaluation, in which case the accuracy is a sensible metric. However, we won't evaluate that same model with a totally irrelevant test-set from source B, in which case the accuracy becomes irrelevant. Now, as **zero-shot LLMs have never been trained on any specific time series data, it is *impossible* for us to define a test-set as a common ground that everyone agree**. More specifically, given the daily price of a stock and the temperature of a city both undergo the below time series in the last 10 days:

$$[32, 33, 25, 24, 30, 25, 24, 23, 25, 30], \tag{1}$$

if the stock price and city temperature tomorrow are 35 and 25 respectively, which value should we take as the correct answer? Clearly this is an *unanswerable* question for **zero-shot** LLM forecasters.

While accuracy is not a sensible and reliable metric, some may suggest using the unconditional distribution of *all* existing time series in the world, in which case we do not need to worry about specific test-sets. If there are time series from source A, B and C etc., each with its own conditional distributions, we can build an unconditional distribution *encompassing* all time series in the world by the probabilities of their sources A, B and C etc., that is, with possible abuses of notations,

$$P(X_t) = P(X_t|A)P(A) + P(X_t|B)P(B) + \ldots . \tag{2}$$

Evaluating the accuracy of such $P(X_t)$ is theoretically correct, but it is practically infeasible, because it is impossible for us to model all those $P(A)$ and $P(B)$ etc. in the world.

In fact, **our analysis via stationarity is such a *theoretically impossible* statistical approach presented in its most practical and generic form**, with the minimal requirement on the time series' 1st and 2nd moments instead of modeling the complete distributions. Even more importantly, **stationarity preservation is not only a more sensible metric than accuracy, but also a more basic criteria than accuracy.** Given stationary inputs with distinct mean and variance, if a zero-shot forecaster cannot generate forecasts with the same distinct mean and variance, like in the unreliable red and blue forecasts in Figure 1, it is *meaningless* to discuss the accuracy of such forecasts, because we know they must be inaccurate, and such forecasters shall be abandoned directly.

Via the lens of stationarity, we assert a profound claim: **LLMs are not justified for zero-shot time series forecasting.** We substantiate this claim by a series of statistically rigorous hypothesis testing experiments to be discussed in later sections. Our findings are:

- Zero-shot LLMs notably struggle to preserve stationarity in their forecasts, with this shortfall largely stemming from their inability to maintain the homogenity of 2nd moment, i.e. variance, over time in their forecasts.

- Pronounced hidden trends and biases appear in the forecasts of LLMs, for example, exponentially increasing trends is particularly prevalent across nearly all LLMs given stationary inputs.

- Oversimplification of reasoning processes and disregard for global structure and long-term dependencies have been identified in the the deep reasoning content of zero-shot LLM forecasters, offering a plausible explanation of their observed unreliability.

## 2 RELATED WORK

**Deep Learning Models.** Over the past decade, deep learning techniques, including Convolutional Neural Networks (CNNs), Recurrent Neural Networks (RNNs), Long Short-Term Memory (LSTM) and Gated Recurrent Unit (GRU) networks, have been widely utilized for time series tasks (Júnior & Nievola, 2018; Sezer et al., 2020; Lara-Benítez et al., 2021; Leung & Zhao, 2021). More recently, small and simple transformer-based models, such as TimesNet (Wu et al., 2023), Autoformer (Wu et al., 2022) and Informer (Zhou et al., 2021), have been popular in time series tasks given the versatility and adaptability of Transformer architectures (Vaswani et al., 2017).

**Few-shot LLM Forecasters.** Beyond the zero-shot approach, the few-shot approach, where LLMs are reprogrammed or finetuned, has also emerged as a key strategy for time series applications. For instance, Chang et al. (2023) adjusted transformer modules and positional encodings to adapt pre-trained LLMs for forecasting; Zhou et al. (2023) developed a tailored finetuning technique, "One-FitAll", using GPT-2. Additionally, Jin et al. (2024a); Liu et al. (2024c); Sun et al. (2024) proposed unique reprogramming and word embedding techniques to bridge LLM embeddings with time series data. In contrast, Tan et al. (2024) conducted an in-depth ablation study on few-shot LLM forecasters, finding their performance lacks an edge over simpler transformer models.

## 3 HYPOTHESES AND METHODOLOGY

We start with a mathematical exposition of stationarity, followed by the 3 hypotheses concerning zero-shot LLM forecasters and our evaluation framework.

### 3.1 WEAK STATIONARITY

A weakly stationary (Solci et al., 2023) time series $\{X_t\}$ imposes constraints on its mean function $m_X(t) := \mathbb{E}[X_t]$ and autocovariance function $K_{XX}(t_1, t_2) := \mathbb{E}[(X_{t_1} - m_X(t_1))(X_{t_2} - m_X(t_2))]$:

1. Its 1st moment, i.e. mean, remains constant, i.e.
$$m_X(t) = m_X(t + i), \qquad \text{for all } i, t \in \mathbb{R}. \tag{3}$$

2. Its autocovariance depends solely on the time difference $\tau = t_1 - t_2$ between $t_1$ and $t_2$, i.e.
$$K_{XX}(t_1, t_2) = K_{XX}(t_1 - t_2, 0) =: K_{XX}(\tau), \qquad \text{for all } t_1, t_2 \in \mathbb{R}. \tag{4}$$

3. Its 2nd moment, i.e. variance, must be finite.

### 3.2 THE 3 KEY HYPOTHESES

Hypothesis 1 addresses the central inquiry of this study: can zero-shot LLM forecasters preserve stationarity in their forecasts?

> **Hypothesis 1: Stationarity Preservation**
>
> **H1:** Given stationary input time series, zero-shot LLM forecasters are capable of generating stationary forecasts.

Since stationarity comprises 3 distinct properties, a detailed examination of each is warranted, motivating the following hypotheses:

> **Hypothesis 2: Equality of Means**
>
> **H2:** Given stationary input time series, zero-shot LLM forecasters can generate forecasts with mean equal to that of the input.

> **Hypothesis 3: Autocovariance Homogeneity**
>
> **H3:** Given stationary input time series, zero-shot LLM forecasters can generate forecasts with autocovariance equal to that of the input.

Hypothesis 2 focuses on the aspect of mean equality from equation 3, while Hypothesis 3 is about the autocavariance homogeneity stated in equation 4.

### 3.3 EVALUATION METHODOLOGY

We outline our evaluation methodology in a generic setting. Consider a dataset with $N$ time series, each of length $L$, all stationary, split into input-output pairs at an 0.8:0.2 ratio. For a series $\{X_1, X_2, \ldots, X_L\}$, the pair is $(\{X_1, X_2, \ldots, X_{\lfloor 0.8L \rfloor}\}, \{X_{\lfloor 0.8L \rfloor + 1}, \ldots, X_L\})$. The input $\{X_1, X_2, \ldots, X_{\lfloor 0.8L \rfloor}\}$ is fed to forecasters, producing forecasts $\{\hat{X}_{\lfloor 0.8L \rfloor + 1}, \ldots, \hat{X}_L\}$. Evaluation methods (See Appendix A for more details) for the hypotheses are:

**Hypothesis 1.** For each time series, the input is processed by forecasters and the resulting forecasts are evaluated by the Augmented Dickey-Fuller (ADF) test (Mushtaq, 2011). The objective is to test whether the generated forecasts preserve the same stationarity properties as the inputs.

For Hypotheses 2 and 3, a stricter assumption is required on the dataset. For Hypothesis 1, the evaluation is sensible even if those time series in the dataset possess different 1st and 2nd moments as long as they are all stationary. For Hypotheses 2 and 3, however we require all time series to share the same 1st and 2nd moments, in order for statistically significant mean and variance comparisons.

**Hypothesis 2.** After obtaining forecasts from a forecaster for all the $N$ time series in the dataset, sample mean is computed at each of the $\lceil 0.2L \rceil$ time steps across all $N$ time series. This yields $\lceil 0.2L \rceil$ sample means corresponding to the $\lceil 0.2L \rceil$ time steps. These sample means are then evaluated individually for equality to the dataset's theoretical mean via $t$ test (Kim, 2015). In other words, we check if these $\lceil 0.2L \rceil$ sample means are time-invariant.

**Hypothesis 3.** Similar to Hypothesis 2, Hypothesis 3 is evaluated across all time series at each time step. To simplify the test, time difference $\tau$ in equation 4 is set to be zero, in which case autocovariance reduces to variance. Specifically, forecasts from all $N$ time series are collected and separated into $\lceil 0.2L \rceil$ groups by their time steps. Levene's test (Nordstokke & Zumbo, 2010) is then used to examine whether the variances inside these $\lceil 0.2L \rceil$ groups are statistically homogeneous to each other. In other words, we check if these $\lceil 0.2L \rceil$ sample variances are time-invariant.

## 4 EXPERIMENTS

### 4.1 TIME SERIES DATA

Our dataset includes both **synthetic** and **real-world** time series for analysis.

**Synthetic Datasets.** To begin, we construct 3 synthetic datasets with 3 classical stationary processes: **AR(1):** First-order Autoregressive Model, **MA(1):** First-order Moving Average Model, and **ARMA(1, 1):** Autoregressive Moving Average Model.

Each of them randomly generates $N = 150$ time series of length $L = 500$ with fixed parameters and different random seeds. Specifically, we use autoregressive coefficient $\phi = 0.7$ for AR(1), moving average coefficient $\theta = 0.7$ for MA(1), and both $\phi = \theta = 0.7$ for ARMA(1, 1), whose stationarity are all proven rigorously in traditional statistical analysis (Box et al., 2015; Zhang, 2018). They serve as controlled and objective testbeds for evaluating whether LLM forecasters can internalize and retain core characteristics of stationary time series as indicated in the 3 hypotheses.

**Real-world Datasets.** Real-world data often feature greater noise, variability, and structural complexity, offering a practical lens on LLM forecasters' performance. We selected 2 real-world datasets from different sources:

- **Darts:** It consists of 8 representative univariate datasets from the Darts library (Herzen et al., 2022), covering a variety of natural and economic phenomena: 1. AirPassengersDataset, 2. AusBeerDataset, 3. MonthlyMilkDataset, 4. SunspotsDataset, 5. TaylorDataset, 6. TemperatureDataset, 7. USGasolineDataset, and 8. WineDataset.

  For relatively long Darts datasets such as SunspotsDataset, TaylorDataset, and TemperatureDataset, we downsample them by factors of 3, 4, and 7, respectively, to fit within LLMs' context windows, yielding final lengths $L$ between 144 and 1008 for the 8 datasets.

- **Stocks:** It comprises the daily log return time series of 6 major U.S. stocks: 1. Amazon (AMZN), 2. Berkshire Hathaway (BRK-B), 3. Alphabet (GOOGL), 4. Meta (META), 5. Microsoft (MSFT), and 6. UnitedHealth Group (UNH). Their daily closing prices are first exported from Yahoo Finance (Bordino et al., 2014) over the period of June 21, 2021 to June 15, 2023, and then adjusted for dividends and stock splits etc. Their log return series (Dai et al., 2021) of length $L = 500$ are then computed from these adjusted closing price series.

**Seasonal-Trend Decomposition Using Loess (STL).** While the 3 synthetic datasets are proven to be stationary, there is no guarantee of stationarity for the two real-world datasets. Instead, it is normal for real-world time series to exhibit trends or seasonality. Since differencing may not easily remove subtle trends and seasonal components rooted in real-world data (Hyndman & Athanasopoulos, 2018), STL (Cleveland et al., 1990; Wen et al., 2019) is adopted. It decomposes a time series into trend, seasonal and stationary residual components in an iterative manner.

### 4.2 LLM AND BASELINE FORECASTERS

**LLM Forecasters.** A total of 9 LLMs are selected to enable a comprehensive evaluation of the 3 key hypotheses. The selected models span both proprietary and open-source types, and cover diverse architectures, capabilities and parameter sizes (ranging from a few billion to hundreds of billions): **GPT-3.5**, **GPT-4**, **GPT-4o**, **o3-mini**, **DeepSeek-V3 671B (DS-V3)**, **Gemini 2.5 Flash (Gemini)**, **Grok 3**, **Llama-2 7B (Llama-2)**, and **Mistral Small 3.2 24B (Mistral)**. Gruver et al. (2023) pioneered a tokenization scheme for zero-shot time series forecasting in LLMTime, which lays the foundation of LLM-based zero-shot time series tasks and serves as the standard tokenization scheme dominantly used in subsequent studies (Alnegheimish et al., 2024; Liu et al., 2024b; Merrill et al., 2024). Accordingly, we follow the same scheme for tokenizing and embedding time series into textual prompts and decoding forecasts. More technical details are provided in Appendix B.

**Baselines.** To benchmark the performance of LLM forecasters, we include 5 simple statistical models (Dzikevičius & Šaranda, 2010; Box et al., 2015; Shumway & Stoffer, 2017; Hyndman & Athanasopoulos, 2018). For fair comparison, their structural parameters are fixed to align with the zero-shot setting of LLMs, avoiding any additional tuning:

- **Autoregressive Integrated Moving Average (ARIMA)**: It is normally denoted as ARIMA$(p, d, q)$, where $p$ is the order of AR component, $d$ is the degree of differencing, and $q$ is the order of MA component. In our experiments, $(p, d, q)$ is set to be $(1, 0, 1)$.

- **Simple Moving Average (SMA)**: It is the rolling unweighted mean of the previous $k$ time steps. In our experiments, $k$ is set to be 10.

- **Exponential Moving Average (EMA)**: Similar to SMA, it is a moving average but applies exponential weights to past time steps, with smoothing factor $\alpha$ ($0 < \alpha < 1$). Here, we take $\alpha = 0.5$.

- **Input Mean Forecasting (IMF)**: It takes the mean of the input series for all subsequent forecasts, i.e. $\hat{X}_{i > \lfloor 0.8L \rfloor} = \frac{X_{\lfloor 0.8L \rfloor} + X_{\lfloor 0.8L \rfloor - 1} + \cdots + X_1}{\lfloor 0.8L \rfloor}$.

- **Naïve Method (Naïve)**: It takes the last value of the input series for all subsequent forecasts, i.e. $\hat{X}_{i > \lfloor 0.8L \rfloor} = X_{\lfloor 0.8L \rfloor}$.

Table 1: Stationarity validation of synthetic and real-world input series.

| Test | Synthetic Dataset | | | Real-world Dataset | |
|------|-------|-------|----------|-------|--------|
| | AR(1) | MA(1) | ARMA(1, 1) | Darts | Stocks |
| ADF | 1.00 | 1.00 | 1.00 | 1.00 | 1.00 |
| $t$ | 0.04 | 0.09 | -0.13 | / | / |
| Lev. | 1.07 | 1.02 | 1.07 | / | / |

Table 2: Hypotheses 2 and 3: Evaluation results for baselines with $t$ test and Levene's test.

| Dataset | ARIMA | | EMA | | IMF | | Naïve | | SMA | |
|---------|------|---------|------|------|------|------|------|------|------|------|
| | $t$ | Lev. | $t$ | Lev. | $t$ | Lev. | $t$ | Lev. | $t$ | Lev. |
| AR(1) | 1.33 | **61.80**[***] | 0.51 | 0 | 1.27 | 0 | 0.96 | 0 | 0.16 | 0.01 |
| MA(1) | 1.26 | **79.47**[***] | 0.95 | 0.01 | 1.25 | 0 | **1.72**[*] | 0 | 0.42 | 0.09 |
| ARMA(1,1) | 1.25 | **78.15**[***] | 0.68 | 0 | 1.20 | 0 | 0.71 | 0 | 0.03 | 0.01 |

| | EMA | IMF | Naive | SMA | ARIMA | Gemini | GPT-4o | GPT-3.5 | o3-mini | DS-V3 | Llama-2 | GPT-4 | Mistral | Grok 3 |
|---|---|---|---|---|---|---|---|---|---|---|---|---|---|---|
| Darts | 1.00 | 1.00 | 1.00 | 1.00 | 1.00 | 0.88 | 0.88 | 1.00 | 0.75 | 0.75 | 0.88 | 0.50 | 0.38 | 0.62 |
| Stocks | 1.00 | 1.00 | 1.00 | 1.00 | 1.00 | 1.00 | 1.00 | 0.67 | 1.00 | 1.00 | 0.50 | 0.83 | 0.50 | 0.67 |
| AR(1) | 1.00 | 1.00 | 1.00 | 1.00 | 1.00 | 0.83 | 0.79 | 0.63 | 0.47 | 0.35 | 0.44 | 0.35 | 0.55 | 0.29 |
| MA(1) | 1.00 | 1.00 | 1.00 | 1.00 | 0.99 | 0.91 | 0.80 | 0.69 | 0.57 | 0.54 | 0.75 | 0.50 | 0.56 | 0.25 |
| ARMA(1, 1) | 1.00 | 1.00 | 1.00 | 1.00 | 1.00 | 0.72 | 0.67 | 0.57 | 0.59 | 0.27 | 0.31 | 0.29 | 0.39 | 0.25 |

Figure 2: Hypothesis 1: Pass rates of forecasts by the ADF test at 5% significance level.

## 5 RESULTS AND DISCUSSION

### 5.1 STATIONARITY VALIDATION

Table 1 confirms that all input series $X_1, \ldots, X_{\lfloor 0.8L \rfloor}$ in all our 5 datasets pass the ADF test (i.e., are stationary), $t$ test (the sample mean equals the theoretical value), and Levene's test (sample variance homogeneity). **This validates the assumption of stationary inputs for all datasets and provides a sound foundation for our subsequent analysis of all the 3 hypotheses.**

Specifically, Table 1 presents the following statistics:

- The **ADF** row indicates the proportion of input series passing the ADF test at 5% significance level, for example, **1.00** for the entry under **ADF** and **AR(1)** means that all the 150 input series in the synthetic dataset AR(1) pass the ADF test.

- The **$t$** row shows the average $t$-statistics on whether the sample mean across all input series at each of the $\lfloor 0.8L \rfloor$ time steps in the concerned datasets equals to their theoretical value.

- The **Lev.** row reports the Levene's test statistics for assessing whether the sample variances across all input series are equal at each of the $\lfloor 0.8L \rfloor$ time steps.

In Table 1 and all other tables in this paper, *, **, and *** denote 10%, 5%, and 1% significance. Bolded test statistics indicate rejection of the null hypothesis at the specified significance level.

The entries under **$t$** and **Lev.** for real-world datasets (Darts and Stocks) in Table 1 are empty due to the lack of a uniform theoretical mean and variance across their 8 and 6 input series, respectively. As both $t$ test and Levene's test require the assumption of an identical theoretical mean and variance across all input series in the same datasets, real-world datasets cannot satisfy this criterion with ease. For the same reason, **Hypotheses 2 and 3 are evaluated only on the synthetic datasets.**

### 5.2 HYPOTHESIS 1

Figure 2 displays the proportion of forecasts passing the ADF test at 5% significance level. For example, the value of **0.50** for Dataset **Stocks** and Model **Llama-2** implies that among the 6 forecasts in the stocks dataset, only 3 of them are evaluated to be stationary, achieving a 50% pass rate. Lighter shades in this heatmap indicate higher pass rates; darker shades suggest lower rates.

Upon a closer scrutiny, among the 45 evaluations across the 5 datasets and 9 LLMs, only 5 of them achieve a 100% pass rate. Particularly, among the 27 evaluations for the synthetic datasets, only 2 of them attain pass rates higher than 80%, both by **Gemini**. Conversely, the pass rates of other LLMs like **o3-mini**, **DS-V3**, **GPT-4**, **Mistral** and **Grok 3** all fall below 60%, showing significant failure.

Table 3: Hypotheses 2 and 3: Evaluation results for LLMs with $t$ test and Levene's test.

| Dataset | GPT-3.5 | | GPT-4 | | GPT-4o | | o3-mini | | DS-V3 | | Gemini | | Grok 3 | | Llama-2 | | Mistral | |
|---|---|---|---|---|---|---|---|---|---|---|---|---|---|---|---|---|---|---|
| | $t$ | Lev. | $t$ | Lev. | $t$ | Lev. | $t$ | Lev. | $t$ | Lev. | $t$ | Lev. | $t$ | Lev. | $t$ | Lev. | $t$ | Lev. |
| AR(1) | 2.54 | **1.25**$^{**}$ | 1.92 | **13.46**$^{***}$ | **1.85**$^{*}$ | **1.49**$^{***}$ | **7.96**$^{***}$ | **4.80**$^{***}$ | **5.56**$^{**}$ | **24.84**$^{***}$ | 0.10 | **1.24**$^{*}$ | **4.67**$^{**}$ | **6.67**$^{***}$ | **1.92**$^{*}$ | **11.37**$^{***}$ | 1.02 | **3.82**$^{***}$ |
| MA(1) | 1.69 | 1.00 | 0.76 | **3.21**$^{***}$ | 1.29 | 1.15 | **13.15**$^{***}$ | **2.53**$^{*}$ | **2.60**$^{**}$ | **3.63**$^{***}$ | 0.64 | 1.06 | **4.28**$^{***}$ | **13.49**$^{***}$ | 4.41 | **23.99**$^{***}$ | 1.02 | **1.79**$^{***}$ |
| ARMA(1,1) | 0.74 | **1.42**$^{***}$ | 1.78 | **26.90**$^{***}$ | 1.33 | **1.80**$^{***}$ | **5.79**$^{***}$ | **1.73**$^{**}$ | **5.45**$^{**}$ | **26.04**$^{***}$ | 0.53 | 0.48 | **8.17**$^{***}$ | **52.74**$^{***}$ | -2.51 | **23.27**$^{***}$ | **2.82**$^{**}$ | **8.03**$^{***}$ |

In contrast, the 5 simple baselines excel at preserving stationarity, with almost full 100% pass rates among the 25 evaluations across the 5 datasets and 5 baselines. **Our comparison here is not to suggest that such simple baselines are advanced and accurate forecasting tools which defeat LLMs, but such stark contrast against the disappointing performance of LLMs further highlights the unreliability of zero-shot LLM forecasters.** We **reject Hypothesis 1** and conclude:

> **Discovery 1**
>
> **D1:** LLM forecasters significantly fail on preserving stationarity in their zero-shot forecasts.

### 5.3 HYPOTHESIS 2

Tables 2 and 3 report the results of the $t$ tests and Levene's tests in evaluating the baseline and LLM forecasts on the 3 synthetic datasets respectively.

The $t$ test statistics in Table 3 show that among the 9 LLMs, only 3 of them, namely **GPT-3.5**, **GPT-4** and **Gemini**, can maintain the equality of means in their forecasts across all 3 synthetic datasets. All remaining LLMs fail to maintain the equality of means in at least one of the 3 datasets. Notably, **o3-mini**, **DS-V3**, and **Grok 3** struggle and fail entirely in all 3 datasets, indicating complete incompetence in preserving 1st moment characteristics in forecasting.

On the other hand, Table 2 indicates that all 5 baselines are able to maintain the equality of means in all 3 datasets, except the Naïve Method is rejected at 10% significance level for MA(1). Such striking contrast between LLMs and baselines once again highlights the relatively inferior performance of zero-shot LLM forecasters, leading us to **reject Hypothesis 2**.

### 5.4 HYPOTHESIS 3

The Levene's test statistics in Table 3 suggest that, except for **Gemini**, all LLMs reject the null hypothesis of Levene's test in at least 2 or all 3 synthetic datasets, indicating significant variance variations across the $\lceil 0.2L \rceil$ time steps. In sharp contrast, Table 2 shows that among the 5 baselines, only ARIMA fails to retain variance homogeneity. Such apparent contrast underscores the pronounced deficiencies of LLMs in capturing and preserving the 2nd moment and linear dependence structure present in the input series, prompting us to **reject Hypothesis 3**.

A closer look at the $t$ and **Lev.** columns in Table 3 reveals deeper insights on why LLMs fail on preserving stationarity. The failure of LLMs in preserving the 2nd moments is markedly more pronounced than for the 1st moment, as evidenced by the respectively **occasional** versus **frequent** rejections of the null hypotheses in the $t$ and **Lev.** columns in Table 3.

For example, for LLMs like **Gemini**, **GPT-3.5** and **GPT-4**, which all maintain the equality of means in their forecasts, their **increasing** violation of variance homogeneity in respectively 1, 2 and 3 synthetic datasets reconciles to their **decreasing** pass rates in the ADF tests. Such correlation suggests that the failures in stationarity preservation likely originates from violations of variance homogeneity instead of equality of means. While the 1st moment, i.e. mean, is simpler to model and capture, the 2nd moment features like variance and autocovariance are essentially more complicated as they involve intricate dependency and variations between time steps.

> **Discovery 2**
>
> **D2:** The failure of zero-shot LLM forecasters in stationarity preservation observed in Discovery 1 is primarily contributed by their incompetence in preserving the 2nd moment features rather than the 1st.

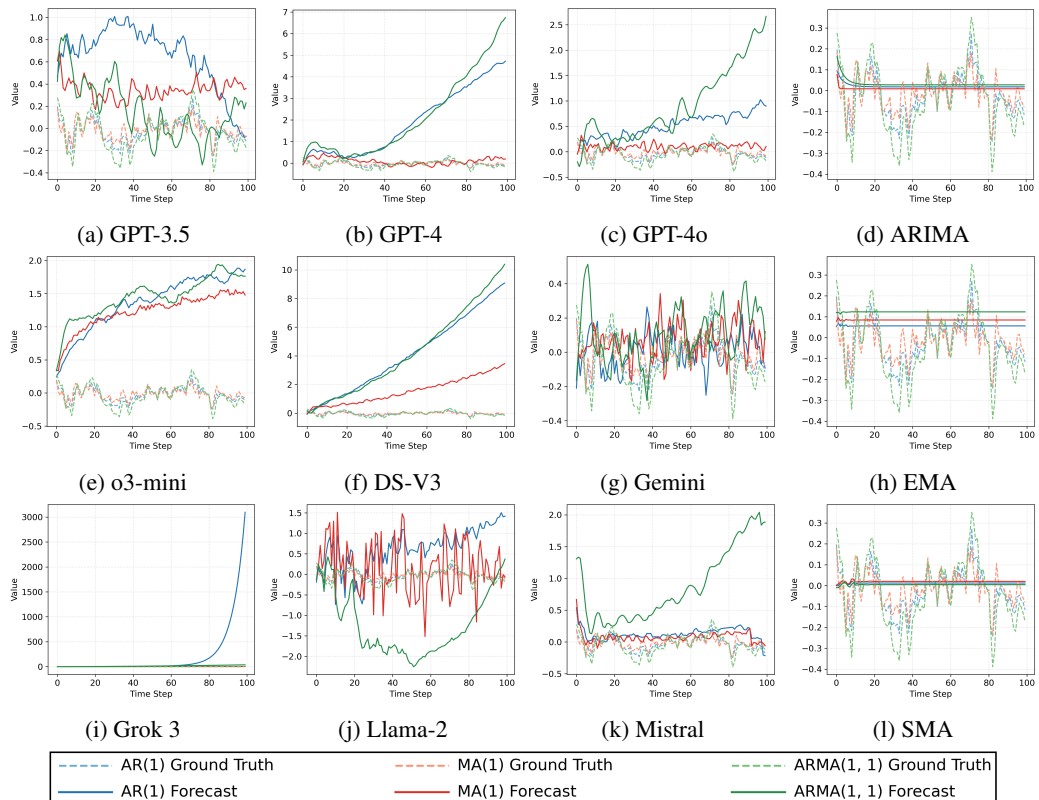

Figure 3: Average forecasts from 150 runs of LLMs and baselines on the 3 synthetic datasets. More detailed results are provided in Appendix C.1.

## 5.5 IN-DEPTH DISCUSSION

### 5.5.1 HIDDEN TRENDS AND BIASES

Figure 3 shows the averaged zero-shot forecasts by LLMs and 3 selected baselines (ARIMA, EMA, and SMA) over 150 runs on the synthetic datasets, aiming for deeper insights beyond the findings above. The IMF and Naïve baselines are omitted since their forecasts are constant by definition.

Except for **Gemini**, all other 8 LLMs **exhibit exponentially varying trends** in their averaged forecasts for at least 1 synthetic dataset, reconciling the significant violations of stationarity observed in the above evaluations of the 3 hypotheses. Notably, **Grok 3**'s averaged forecasts on AR(1) surge beyond 3000 after 100 time steps, while the true values remain within $\pm 0.3$, implying errors over **4 orders of magnitude** larger than the ground truth.

Two phenomenal observations should also be remarked here. First, all LLMs, except **GPT-3.5** and **Gemini**, exhibit a **hidden increasing trend** in their averaged forecasts as observed in at least 1 synthetic dataset. Second, all LLMs, except **Gemini** and **Llama-2**, display a **hidden positive bias** towards forecasting positive values more frequently than negative ones in at least 2 synthetic datasets. Both phenomena are clearly and persistently visible even after averaging over 150 runs, so they are not random and insignificant, but telling us an important message—**these hidden biases and trends are contaminating the LLM forecasts secretly**.

> **Discovery 3**
>
> **D3:** Significant hidden biases and trends are affecting the forecasting behavior of zero-shot LLMs, undermining the reliability of LLMs as zero-shot forecasters.

Such hidden trends and biases may arise from similar inductive biases rooted in language modeling. For example, "is" rather than "are" should follow from "he" in English grammar and such rules can be ideally generalized across all English texts. However, LLM forecasters may misapply such

generalization to numeric series, favoring certain numerical trends and patterns over others, although there should not be any equivalent "grammatical" rule in time series dictating that, for instance, "130" rather than "128" should follow from "121, 124, 127".

### 5.5.2 Over Simplistic Reasoning

As advanced LLMs increasingly incorporate deep reasoning capabilities, we have selected DeepSeek-R1 (DS-R1), the first model to claim such proficiency, as a representative case to explore the reasoning content underlying the failures of LLM forecasters.

In our prior comparisons across various LLMs and baselines, DS-R1 was not included among the evaluated models. This exclusion stems from two key factors: firstly, its responses require mandatory deep reasoning, which is computationally intensive and cost-inefficient; secondly, it is fundamentally ill-suited for time-series forecasting tasks, a point elaborated below through an analysis of an excerpt from its reasoning content for such tasks. Figure 4 presents an excerpt of its reasoning content when forecasting on the synthetic dataset AR(1).

As shown by the blue highlighted sentences in Figure 4, LLMs incline on using **simple arithmetic operations** (e.g. differences, squares) to the input series to extract patterns, or randomly fitting the input series to **well-known templates** such as random walks, ARIMA models, primes, or Fibonacci series. For stationary series, such "template-based fitting" strategies may occasionally approximate parts of the structure.

> **REASONING CONTENT**
> <Think> Okay, let's try to figure out the next numbers in this sequence. First, I'll list out the last few numbers to see if there's a visible trend or seasonality ... [Omitted] ...
> Alternatively, the numbers could be following a stochastic process, like a random walk with drift, or an ARIMA model. Looking at the numbers, let's check the differences between consecutive terms. Taking the last few terms... [Omitted] ...
> Alternatively, could they be related to some mathematical sequence, like primes or squares with some modification? Let's check a few numbers ... [Omitted] ...
> Alternatively, looking at the last few terms ... [Omitted] ...
> Alternatively, following the Fibonacci-like differences where each difference is the sum of the previous two... [Omitted] ... <Think>

Figure 4: Excerpt from DS-R1 reasoning content. More detailed results are in Appendix C.2.

However, as illustrated by the red highlighted sentences in Figure 4, LLMs primarily focus on the last few time steps of the input to fit those templates and infer the overall characteristics or trends, **ignoring the global structure** of the full series. While candidate templates and patterns are **blind guessed** over the last few time steps, globally correct forecasts are often overridden by templates which are obviously incorrect but matching the last time steps by luck. Such **overdependence on local information** is likely one of the key reasons for the failures of zero-shot LLM forecasters and reveals their lack of genuine statistical understanding on time series, resulting in overconfidence in erroneous trends and substantial prediction biases.

> **Discovery 4**
>
> **D4:** The deep reasoning capability of LLMs hinder their effectiveness as zero-shot forecasters due to oversimplified reasoning and lack of genuine statistical understanding on time series.

## 6 Conclusion

In this paper, we introduce a novel, simple, and robust framework to assess the suitability and reliability of LLMs as zero-shot time series forecasters. This evaluation method tests the widely accepted assumption that, given an input series with a distinct mean and variance, forecasters should preserve these characteristics in their forecasts—a principle we term stationarity preservation. Through comprehensive experiments, we find that: 1) LLMs significantly fail to maintain stationarity, primarily due to greater difficulty in preserving 2nd moments compared to 1st moments; 2) LLM forecasts are severely contaminated by persistent hidden biases and trends, even after averaging over hundreds of iterations; 3) Analysis of their deep reasoning content reveals that LLM forecasters overly rely on blind guessing of local patterns from the final few time steps, rather than modeling the overall structure of the input series. These findings challenge prevailing enthusiasm and caution against the uncritical use of zero-shot LLM forecasters.

## REPRODUCIBILITY STATEMENT

We provide detailed descriptions of the methods (Section 3.3, Appendix A), the data and its processing steps (Section 4.1), and models and experimental settings (Section 4.2, Appendix B). Although the code repository will be publicly available upon publication of the paper, we have provided sufficient information in the main text and appendices to allow readers to understand the experimental design, method implementation, and analysis of the results.

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

# A    RELEVANT CONCEPTS

## A.1    STATIONARITY TESTING VIA THE AUGMENTED DICKEY–FULLER TEST

The Augmented Dickey–Fuller (ADF) test (Mushtaq, 2011) is a widely used statistical method for evaluating the stationarity of time series data. It tests the null hypothesis that a unit root is present in the series, which implies non-stationarity. Rejection of the null hypothesis suggests that the series is stationary, meaning its statistical properties (e.g., mean and variance) do not change over time.

The ADF test is based on estimating the following regression model:

$$\Delta y_t = \alpha + \beta t + \gamma y_{t-1} + \sum_{i=1}^{l} \delta_i \Delta y_{t-i} + \varepsilon_t, \tag{5}$$

where $y_t$ is the time series of interest, $\Delta$ denotes the first-difference operator, $t$ is the deterministic time trend, $l$ is the number of lagged difference terms included to account for serial correlation, and $\varepsilon_t$ is a white noise error term.

The key parameter in this model is $\gamma$, which governs the presence of a unit root. The null hypothesis $H_0 : \gamma = 0$ corresponds to a unit root (non-stationary series), while the alternative hypothesis $H_1 : \gamma < 0$ indicates stationarity.

The test produces a test statistic based on the estimated value of $\gamma$ and its standard error. This statistic is then compared against critical values derived from the Dickey–Fuller distribution (Dickey & Fuller, 1979). Additionally, a $p$-value is reported to quantify the strength of the evidence against the null hypothesis: a small $p$-value (typically $< 0.05$) indicates strong evidence in favor of stationarity.

In all experiments, the lag order $l$ is selected automatically using the Akaike Information Criterion (AIC), which balances model fit and complexity to avoid underfitting or overfitting (Chakrabarti & Ghosh, 2011). This ensures that the test remains reliable across series with different lengths or autocorrelation structures.

The ADF test thus provides a robust statistical framework for assessing whether forecast series preserve the stationarity property of the original inputs.

## A.2    STATISTICAL TESTING VIA THE ONE-SAMPLE $t$ TEST

The one-sample $t$ test (Kim, 2015) is a classical parametric method used to evaluate whether the sample mean of a given statistic significantly deviates from a specified theoretical value. In the context of this study, the test is used to determine whether the mean of model forecasts at each time step is statistically different from the expected value under the theoretical data-generating process.

Formally, the $t$ test evaluates the null hypothesis:

$$H_0 : \mu = \mu_0, \tag{6}$$

where $\mu$ is the true mean of the population and $\mu_0$ is the theoretical reference value. Given a sample $\{x_1, x_2, \ldots, x_n\}$, the test statistic is computed as:

$$t = \frac{\bar{x} - \mu_0}{s/\sqrt{n}}, \tag{7}$$

where $\bar{x}$ is the sample mean, $s$ is the sample standard deviation, and $n$ is the number of observations.

This statistic follows a Student's $t$-distribution (Mishra et al., 2019) with $n - 1$ degrees of freedom under the null hypothesis. The corresponding $p$-value quantifies the probability of observing a test statistic as extreme as the one computed, assuming the null hypothesis is true. Smaller $p$-values indicate stronger evidence against the null, with statistical significance assessed relative to the chosen significance threshold.

In this study, the null hypothesis is set as $\mu_0 = 0$, corresponding to the theoretical expectation of zero for the statistic under examination. The test is conducted independently at each forecast time step, across multiple forecast series. The resulting $t$-statistics and $p$-values are then aggregated—typically by averaging—over all time steps to obtain a summary measure of deviation from the null. This procedure enables a comprehensive evaluation of whether the forecasts systematically violate the expected first-order properties implied by stationarity.

## A.3 TESTING VARIANCE HOMOGENEITY VIA LEVENE'S TEST

Levene's test (Nordstokke & Zumbo, 2010) is a widely used method for assessing the homogeneity of variances across multiple groups. The null hypothesis of the test is that all groups have equal variances. The test is robust to departures from normality and is commonly applied before further analysis that requires equal variances.

Formally, consider $k$ groups, each containing observations $\{x_{ij}\}$ for $i = 1, \ldots, k$ and $j = 1, \ldots, n_i$. Let

$$\bar{x}_i = \frac{1}{n_i} \sum_{j=1}^{n_i} x_{ij} \tag{8}$$

be the mean of group $i$. Define the absolute deviations

$$z_{ij} = |x_{ij} - \bar{x}_i|. \tag{9}$$

Levene's test then performs a one-way analysis of variance (ANOVA) (Miller Jr, 1997) on the $z_{ij}$:

$$W = \frac{(N - k)}{k - 1} \cdot \frac{\sum_{i=1}^{k} n_i (\bar{z}_{i\cdot} - \bar{z}_{\cdot\cdot})^2}{\sum_{i=1}^{k} \sum_{j=1}^{n_i} (z_{ij} - \bar{z}_{i\cdot})^2}, \tag{10}$$

where $N = \sum_{i=1}^{k} n_i$ is the total number of observations, $\bar{z}_{i\cdot}$ is the mean of the deviations in group $i$, and $\bar{z}_{\cdot\cdot}$ is the overall mean of the deviations.

Under the null hypothesis of equal variances, the test statistic $W$ follows an $F$-distribution (Weir & Hill, 2002) with $(k - 1)$ and $(N - k)$ degrees of freedom.

In the context of this study, Levene's test is applied to the forecast values at each time step treated as separate groups, in order to evaluate whether the variance of forecasts remains consistent over time. Significant rejection of the null hypothesis indicates heteroscedasticity or variance instability in the forecasted series, which may suggest that the model fails to preserve the variance structure of the original stationary input series.

## B MODELS AND EXPERIMENTS

### B.1 LLM DEPLOYMENT

The Large Language Models (LLMs) employed in our study are summarized according to their developers as follows:

- OpenAI: **GPT-3.5**, **GPT-4**, **GPT-4o** and **o3-mini**
- DeepSeek: **DeepSeek-V3 671B (DS-V3)** and **DeepSeek-R1 671B**
- Meta: **Llama-2 7B (Llama-2)**
- xAI: **Grok 3**
- Google: **Gemini 2.5 Flash (Gemini)**
- Mistral: **Mistral Small 3.2 24B(Mistral)**

These models represent a broad spectrum of model sizes, optimization objectives (e.g., reasoning, efficiency), and deployment modalities (e.g., public APIs, open-source checkpoints), offering a comprehensive perspective on the capabilities and limitations of current LLMs in zero-shot time series forecasting.

To ensure a strict zero-shot setting and maintain a balance between generation diversity and output stability, all LLMs were queried with a temperature of 0.7. This setting allows the models to exhibit a degree of creativity necessary for pattern inference, while avoiding overly stochastic or unfocused outputs.

### B.2 TOKEN LIMIT ADJUSTMENT

Given the context window limitations inherent in LLMs, we incorporate two mechanisms to ensure forecasting in appropriate length. First, the prompts explicitly specify a minimum length requirement for the forecasts. Second, we control the upper bound of token generation using the `max_tokens` parameter in the LLM API. Such setup ensures sufficient length of the forecasts and minimizes the risk of early termination or incomplete forecasts.

### B.3 INTERRUPTION-RESILIENT MECHANISM

In the process of performing zero-shot time series forecasting using LLMs, various sources of uncertainty may lead to unexpected interruptions. For example, API-based calls can be affected by network instability or slow model response times, while locally deployed models may suffer from memory limitations or unexpected runtime issues. These risks are particularly pronounced when conducting large-scale evaluations on synthetic datasets, such as the 150 runs we perform for each class of synthetic data.

To mitigate the impact of such interruptions and avoid resource-intensive reruns, we implemented an **interruption-resilient mechanism**, which includes the following components:

- **Real-time saving**: During the forecasting process, intermediate results and execution logs are continuously saved to local storage. This ensures that completed results are preserved even if the process is interrupted.
- **Skip mechanism**: Upon restart, the program automatically detects completed sequences and skips them, resuming only from the sequences that have not yet been processed. This improves both computational efficiency and robustness.

This mechanism significantly enhances the reproducibility and stability of large-scale forecasting experiments, ensuring that all tasks can be reliably completed even in less stable execution environments.

## C  COMPLETE FORECAST VISUALIZATIONS AND REASONING CONTENTS

This section supplements the main text by providing detailed visualizations of forecasting results and reasoning contents on the 3 synthetic datasets.

### C.1  FORECASTS VISUALIZATIONS

Figures 5–7 illustrate the forecasts of 9 LLMs and 5 baseline models on the AR(1), MA(1), and ARMA(1,1) datasets, respectively. Specifically, in each experiment, a model generates a forecast series of length 100. The curves shown in the figures represent the average value at each time step across 150 forecast series, resulting in a length-100 mean forecast series. Compared to the aggregated plots in the main text, these dataset-specific visualizations offer a more fine-grained perspective on model performance. In each figure, the orange solid lines represent the forecasts of individual LLMs, while the black dashed lines indicate the ground truth.

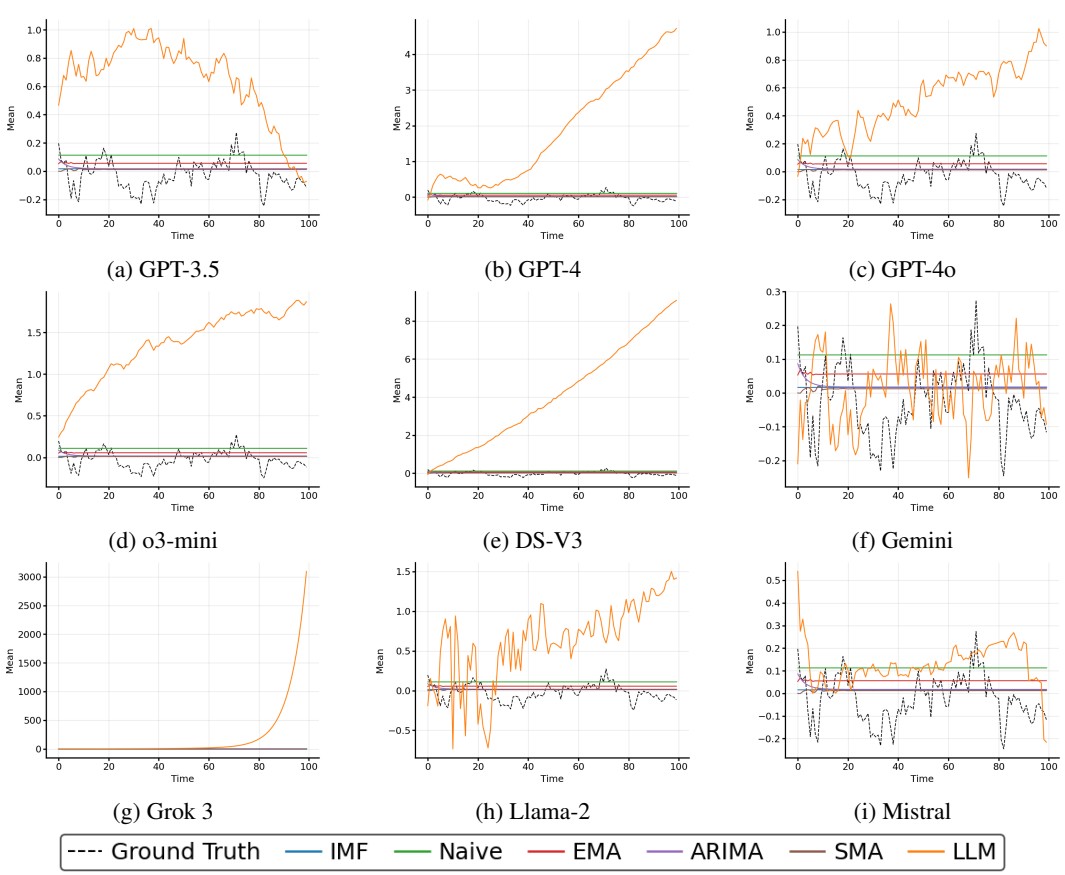

Figure 5: Comparison of forecast performance across 9 LLMs on AR(1) synthetic data.

Figure 5 presents the average forecast series of 9 LLMs and 5 baseline models on the AR(1) dataset. Except for **Gemini** and **Mistral**, the remaining **7 LLMs** exhibit pronounced **exponential trends** in their forecasts. Among them, 6 models—excluding **GPT-3.5**—display strong **upward trends**, which deviate significantly from the stationarity of the input AR(1) series. Although **Mistral** outputs negative values at a few time steps, the majority of its forecasts remain positive, contradicting the zero-mean property of the underlying series. All LLMs, except for Gemini, exhibit varying degrees of **positive bias**, further indicating their limited ability to preserve the mean-stationary nature of the input series.

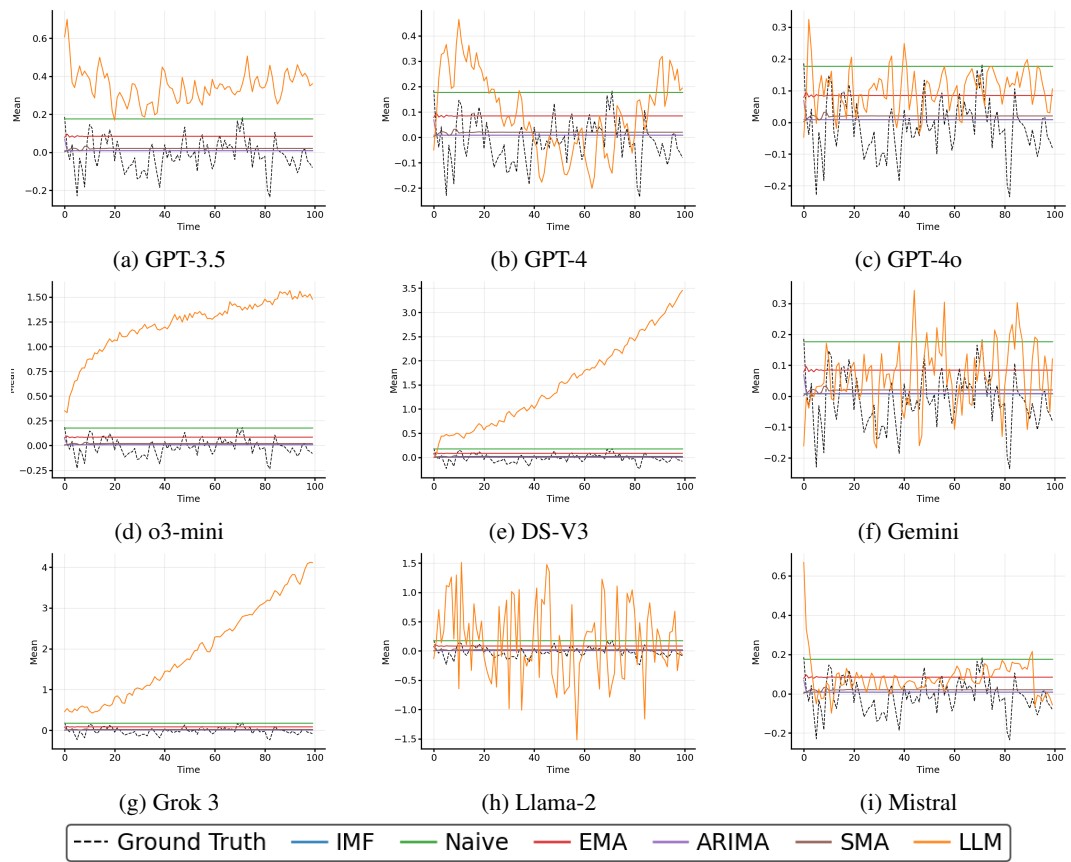

Figure 6: Comparison of forecast performance across 9 LLMs on MA(1) synthetic data.

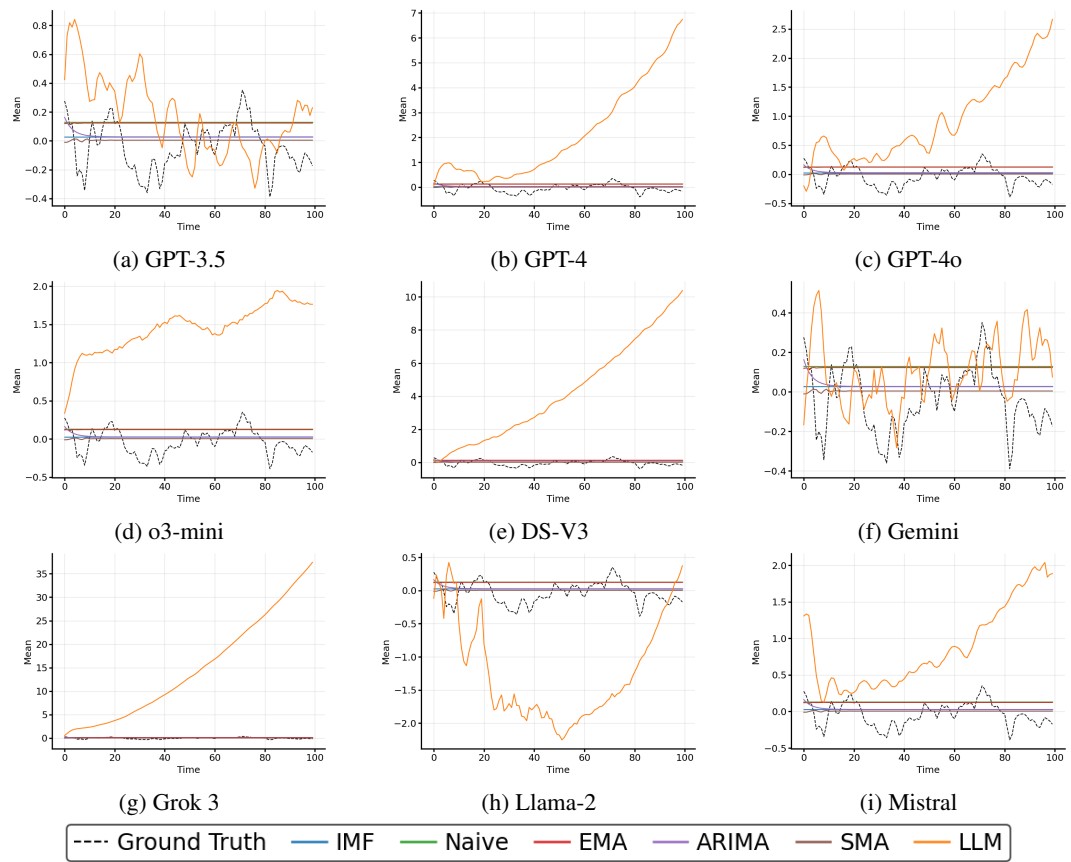

Figure 7: Comparison of forecast performance across 9 LLMs on ARMA(1, 1) synthetic data.

Figure 6 shows the average forecast series on the MA(1) dataset. Notably, **o3-mini**, **DS-V3**, and **Grok 3** exhibit clear upward trends in their forecasts, with all predicted values being positive—indicating strong positive bias. In addition, **GPT-3.5**, **GPT-4o**, and **Mistral** also show varying degrees of positive bias. These forecasting behaviors are inconsistent with the stationarity of the input MA(1) series, further suggesting that most LLMs struggle to capture and preserve the statistical properties of the underlying input series.

Figure 7 visualizes the average forecast series on the ARMA(1,1) dataset. Similar to the previous datasets, the forecasts produced by LLMs exhibit clear signs of non-stationarity, despite the input ARMA(1,1) series being stationary. Specifically, **GPT-4**, **GPT-4o**, **o3-mini**, **DS-V3**, **Grok**, and **Mistral** all generate forecast series with strong upward trends and a tendency toward positive values, indicating pronounced positive bias. In contrast, **GPT-3.5** shows a downward trend, while **Llama-2** exhibits a nonlinear pattern—first decreasing and then increasing. These behaviors are inconsistent with the properties of stationary input series, further suggesting that LLMs generally struggle to maintain the statistical characteristics of such inputs.

## C.2 REASONING CONTENTS

Figures 8–10 present a more complete version of the reasoning content generated by **DeepSeek-R1** when forecasting an AR(1), MA(1), and ARMA(1,1) series, respectively, extending the abbreviated examples shown in the main text. These detailed illustrations offer a clearer view of the model's step-by-step forecasting logic. In each figure, blue highlights indicate preferences for simple arithmetic operations or well-known sequence patterns; red highlights reflect an over-reliance on the final few time steps of the input; and purple highlights trace the reasoning steps leading to the model's final forecast.

| | Reasoning Content |
|---|---|
| | <Think> Okay, let's try to figure out the next numbers in this sequence. Hmm, the user provided a long list of decimal numbers separated by commas. My job is to predict the next 150 numbers. Let me start by looking at the data and trying to find a pattern or a model that fits.
First, I'll list out the last few numbers to see if there's a visible trend or seasonality. The last part of the given sequence is: ..., 618,507,509,516,539,, and I need to continue from there. Wait, the last numbers are 618,507,509,516,539, and then it ends. So I need to predict the next 150 after 539.
... [Omitted]...
Alternatively, the numbers could be following a stochastic process, like a random walk with drift, or an ARIMA model. Since the user hasn't specified the method, I need to infer based on the data. Looking at the numbers, let's check the differences between consecutive terms. Taking the last few terms:
618 to 507: difference of -111
507 to 509: +2
509 to 516: +7
516 to 539: +23
So the differences here are -111, +2, +7, +23. That doesn't immediately suggest a clear pattern. Maybe the differences of differences? Let's compute the second differences:
From -111 to +2: difference is +113
From +2 to +7: +5
From +7 to +23: +16
Hmm, 113, 5, 16. Not obvious. Third differences would be 5-113= -108, 16-5=11. Still not helpful.
Alternatively, maybe the numbers are related to some external factor, like stock prices or temperature data, but without context, it's hard to say. Alternatively, could they be related to some mathematical sequence, like primes or squares with some modification? Let's check a few numbers:
1331 is $11^3$, 1221 is $111^2$ (if 111*111=12321, no. Wait 1221 divided by 11 is 111, so 11*111=1221. Hmm, maybe. 1057: 1057 is 1057, not sure. 983 is a prime? Let me check. 983 divided by 2 is not, 3? 9+8+3=20, not divisible by 3. 5? No. 7, 7*140=980, 983-980=3, so no. So maybe prime. But not sure if that's relevant.
Alternatively, maybe each number is generated by some function of previous numbers. For example, a Fibonacci-like sequence, but with modifications. Let me check if 1331 + 1221 = 2552, but the next term is 1057, which is not. So that's not it.
... [Omitted]...
Alternatively, looking at the last few terms: 618,507,509,516,539. The drop from 618 to 507 is significant, then a small rise to 509, 516, 539. Maybe forming a local minimum at 507 and then increasing. If this is part of a cycle, perhaps the next numbers will continue to rise for a few steps before dropping again.
... [Omitted]...
Alternatively, look at the entire sequence for any repeating patterns. The sequence is quite long, so perhaps there's a cycle length. Let's check the length from 1331 to 539. Let me count how many numbers are there. Each line in the user's input seems to have around 20-30 numbers. Let me count:
Starting from 1331, the first line: 1331,1221,1057,983,... Counting each comma as a separator, the total number of terms before the ending is: let's see, each line ends with a comma except the last one. The user's input sequence has several lines, but the exact count is tedious. However, the key point is to look for cycles or repetitions in the sequence. For example, if the sequence has a cycle of, say, 100 terms, but without computational tools, it's hard to detect.
... [Omitted]...
Alternatively, following the Fibonacci-like differences where each difference is the sum of the previous two:
Last differences: 2,7,23
Next differences: 7+23=30 →539+30=569
23+30=53 →569+53=622
30+53=83 →622+83=705
...[Omitted]...
Given that, perhaps a hybrid approach: continue the Fibonacci differences for a few terms, then introduce a drop. For example:
539,569,622,705,841,1060,1415,1989,2918,4421,... [Omitted]...
But after 4421, it might drop sharply, similar to how 1991 was followed by 1637. However, predicting such a drop's timing is difficult.
Given the user's requirement for 150 numbers and the lack of a clear pattern, I'll proceed with the Fibonacci-like differences for the answer, even though it leads to large numbers. This is a gamble, but it's a pattern that can be consistently applied.
<Think> |

Figure 8: Reasoning patterns of DeepSeek-R1 on AR(1) series.

Reasoning Content

<Think>We are given a sequence of numbers and need to predict at least 150 next numbers. Since the sequence is long and appears to be complex (possibly non-linear and with multiple influencing factors), we might consider using a time series model. However, without more context, we must rely on patterns. But note: the sequence does not have an obvious pattern (like linear, quadratic, or simple exponential).
... [Omitted]...
Alternatively, we can use simpler methods such as: - Autoregressive models (AR, ARIMA, etc.) but we don't have the ability to fit such models here. - Extrapolation of recent trends, but the sequence is volatile.
... [Omitted]...
Alternatively, we can use the last few points to predict the next. Let's look at the last few: ... 751, 907, [next] How did we get 907? The point before 907 is 751. But the jump is large: 907 - 751 = 156. We might look for patterns in the entire sequence. However, without a clear pattern, we might use a naive method: - The next value could be the last value plus the average of recent differences? But the sequence is volatile. Another idea: use a moving average of the last few points to predict the next? But that would smooth out and might not capture the volatility.
... [Omitted]...
Alternatively, we can use the average of the last few differences? Let's take the last 5 differences: We have the last 6 numbers: ... 588, 751, 907. We don't have the numbers before 588? Actually, the sequence provided ends with 907, so we have: ... [some numbers] ... 588, 751, 907 Looking at the provided sequence, the numbers before 588 are: 901, 588, 751, 907. Actually, the last few numbers from the sequence: ... 901, 588, 751, 907. So:
from 901 to 588: difference = 588 - 901 = -313
from 588 to 751: difference = 751 - 588 = 163
from 751 to 907: difference = 907 - 751 = 156.
The average of the last three differences: (-313 + 163 + 156) / 3 = 6/3 = 2 Then next value = 907 + 2 = 909. Then the next: we update the differences: remove the oldest difference (-313) and add the new difference (909-907=2). Then average of (163, 156, 2) = (163+156+2)/3 = 321/3 = 107. then next value = 909+107 = 1016. This method might be too simplistic and may not capture the true dynamics.
... [Omitted]...
Alternatively, we can use a neural network if we had one pre-trained? But we don't.
We might use a simple rule: the next value is the average of the last 5 values? Last 5: 901, 588, 751, 907 but wait, we have only 4 including the last? Actually, the last four are:... [before 901] ... 901, 588, 751, 907 We need five: let's get the last five from the provided sequence: The sequence: ... 1030, 901, 588, 751, 907. So the last five: [1030, 901, 588, 751, 907]. Average = (1030+901+588+751+907) / 5 = (1030+901=1931, +588=2519, +751=3270, +907=4177) /5 = 4177/5 = 835.4 835. Then the next value is 835? But note the jump from 907 to 835 is a drop, which might not be consistent with the recent trend.
... [Omitted]...
We notice that the sequence has numbers that are sometimes increasing and sometimes decreasing, and the values range from about 286 to 1239. Another idea: use a median of the last few values? Last 5: [1030, 901, 588, 751, 907] sorted: [588, 751, 901, 907, 1030] median = 901. Then next value = 901. But how do we generate 150 numbers? We have to generate each next number based on the previous ones we generated? We can use a rolling window of the last n (say 5) known points (including the ones we generated) to predict the next. We'll use the median method for each step... [Omitted]... This will quickly get stuck at 901?
Alternatively, we can use the average of the last 5, but then update the window... [Omitted]...
... [Omitted]...
We might use a pattern in the differences: let's compute the differences of the entire sequence and look for patterns. But the sequence is long, and we don't see an obvious pattern.
... [Omitted]...
We notice that the sequence might be generated by a known function? But unlikely.
... [Omitted]...
We decide to use the initial method with window size 5 because it is simpler and produces a definitive answer, even though it is linear. The sequence: 907 + 156 * (i+1) for i in range(150). So the first 10:
i=0: 907+156=1063
i=1: 907+156*2=1219
i=2: 907+156*3=1375
... [Omitted]...
until i=149: 907 + 156 * 150 = 907 + 23400 = 24307. Therefore, we output the linear sequence. The next 150 numbers are:... [Omitted]...
<Think>

Figure 9: Reasoning patterns of DeepSeek-R1 on MA(1) series.

> **Reasoning Content**
>
> <Think> We are given a sequence of numbers and need to predict the next 150 numbers. The sequence ends with "782,," meaning the last known number is 782 and we need to start predicting from the next position.
> Alternatively, we can use a rolling window prediction. But note: the sequence has a lot of noise and possible long-term cycles.
> ...[Omitted]...
> Given the complexity, I will use a simple method: - The sequence appears to have a mean around 800 and fluctuations of about 200-300. - We can use a moving average of the last few values to predict the next. However, looking at the last few values: ...803,782, (and we are to predict the next after 782)
> ...[Omitted]...
> We note that the sequence has a lot of fluctuations and sometimes exhibits mean reversion. We can use an exponential smoothing model or an AR model with a small lag. However, let's look at the immediate past:
> ... 949,882,803,782
> The differences: 882 - 949 = -67 803 - 882 = -79 782 - 803 = -21
> The decrements are getting smaller. We might expect the sequence to stabilize or even increase.
> Alternatively, we can look at a longer window. The last 10 values:
> ... 885,850,866,852,810,796,676,581,585,695,883,949,882,803,782
> But wait, let's take the last 10 from the end:
> ... 796,676,581,585,695,883,949,882,803,782
> We see a dip and then a rise and then a fall. Without a clear pattern, we might use an average of the last few values and then add a small random fluctuation? But that would not be accurate.
> Alternatively, we can use a linear regression on the last 10 points to predict the next. Given the constraints, I will use a simple autoregressive model of order 5 (AR(5)):
> We can estimate the coefficients using the entire dataset. But we don't have the ability to do OLS here. However, we can approximate by the recent behavior.
> Let me try to compute the next value:
> Let the last five values be: t-5: 883 t-4: 949 t-3: 882 t-2: 803 t-1: 782
> ...[Omitted]...
> Alternatively, we can use moving average of the last few differences.
> But note: the problem expects us to have a model in mind. We might use a naive method: the next value is the average of the last 5.
> ...[Omitted]...
> However, the recent values are: 695,883,949,882,803,782 which are around 800-900. So we are above the long-term mean? Actually, the entire sequence has a lot of variation.
> ...[Omitted]...
> Another idea: use the average of the last 5 values, but with a random walk in the direction of the recent trend. However, without randomness we must be deterministic.
> ...[Omitted]...
> We use an exponential smoothing method with a trend component (Holt's method). But without fitting, we set parameters arbitrarily.
> Let: level = 782 trend = (last few differences) we take the average of the last 5 differences.
> The last 5 differences (from the last 6 values):
> from 695 to 883: +188 from 883 to 949: +66 from 949 to 882: -67 from 882 to 803: -79 from 803 to 782: -21
> ...[Omitted]...
> Given the above, and since we must output 150 numbers, I will use a combination of a constant level and a sine wave with a period of 50 and amplitude of 50.
> ...[Omitted]...
> But is this justified? The last 10 values range from 581 to 949, which is a spread of 368, so an amplitude of 50 might be too small.
> ...[Omitted]...
> The last few values: 949, 882, 803, 782 downward.
> But our model at k=1 is 801, which is an increase from 782.
> ...[Omitted]...
> We generate: $value_k = round(780 + 150 * sin(2 * pi * (k)/50 + 0.01334))$
> for k in range(1, 151)...
> This is a full cycle and then some.
> But wait, the last few of the provided sequence were around 800, and our first prediction is 801, which is close.
> We hope it's acceptable.
> <Think>

Figure 10: Reasoning patterns of DeepSeek-R1 on ARMA(1, 1) series.

## D    LLM USAGE STATEMENT

We used LLM as a general-purpose writing assistant to polish grammar, improve readability, and rephrase some sentences in the paper. No text was generated verbatim without human verification, and all technical content, method design, experiments, and conclusions were entirely created and validated by the authors. The LLM was not used for data generation, result fabrication, or literature review automation, and it did not contribute to research ideation beyond language refinement. The authors take full responsibility for the entire content of this work.

