# OpenReview forum: "Are Large Language Models Really Reliable Zero-shot Time Series Forecaster? Failure Analysis via the Lens of Stationarity"
_ICLR.cc/2026/Conference — Submitted to ICLR 2026_

### Official Review · Reviewer_62uq · 2025-10-18

**Soundness:** 3
**Presentation:** 3
**Contribution:** 2
**Rating:** 2
**Confidence:** 4

**Summary:**

The paper argues that large language models (LLMs) are unreliable zero-shot time series forecasters when judged through the basic requirement of stationarity preservation. It proposes a simple framework: given stationary inputs, forecasts should retain the input’s mean and variance, tested using the Augmented Dickey–Fuller test (stationarity), one-sample t tests (mean equality), and Levene’s test (variance homogeneity). Across three synthetic stationary datasets (AR(1), MA(1), ARMA(1,1)) and two real-world collections (after STL detrending), nine popular LLMs frequently fail these criteria, particularly Levene’s test, while simple statistical baselines (e.g., SMA/EMA/IMF and ARIMA with fixed parameters) largely preserve stationarity. The authors find that exponential drift and positive bias persist even after averaging many runs, and they trace the failures primarily to an inability to maintain second-moment structure rather than the mean. An analysis of "reasoning" outputs (e.g., DeepSeek-R1) suggests overreliance on simplistic, template-like pattern guessing from the last few points, with little grasp of global dependence.

**Strengths:**

I like the motivation for the paper. Since pretrained models are now being used for time series forecasting, it makes sense to ask whether these models can maintain an invariant distribution over long forecast horizons.

The experiments are well situated. The experimental question is clear, multiple appropriate metrics are considered, any many foundation models are tested. For example, to minimize confounds, the authors first verify that every input segment is stationary (ADF pass rate = 1.00 across all five datasets) and use synthetic AR(1)/MA(1)/ARMA(1,1) generators with known moment structure to isolate whether models truly preserve first- and second-moment properties. They average results over 150 series per synthetic set with an interruption-resilient resume routine.

The paper does a good job exploring the mechanistic origin of nonstationarity in time series foundation models.
It shows that Augmented Dickey-Fuller failures co-move with Levene’s rejections across datasets, pinning the problem on second-moment (variance) instability rather than mean drift. An audit of DeepSeek-R1’s reasoning traces reveals "template-guessing" from the last few points and neglect of global structure, a plausible mechanism for global nonstationarity.

**Weaknesses:**

I do not think the novelty or scope meets the bar for ICLR. Running ADF or Levene tests on LLM outputs is relatively simple experiment, and the authors consider limited synthetic datasets as well as two real-world datasets. I don’t think it’s surprising that LLM forecast models might exhibit some non-stationarity over time. I would expect that fully-trained contemporary time-series models (e.g., N-BEATS, Patch-TST) exhibit similar drift effects, which I suspect are inevitable in any fixed-precision model over a long enough time horizon. As a result, I do not necessarily agree with the authors’ premise that nonstationarity implies misalignment.

The experimental setting I find a bit odd. The authors use well-known LLMs to forecast time series. They convert time series into language using the LLMTime encoding scheme. This is not a common evaluation setting, and most forecasting practitioners are currently using zero-shot time series forecasting models (Chronos, Times-FM, Moirai) for forecasting, rather than chat models. It is not clear to me what the analog of nonstationarity would be for language, and so language models violating stationarity assumptions does not necessarily imply that something has gone wrong, since time series are not their intended use case.

The writing editorializes claims, with bold phrases such as "these hidden biases and trends are contaminating the LLM forecasts secretly" as well as highlighted "Discoveries" supported by limited experiments. This is overselling, particularly given the modest experimental scope. Even with stronger experiments, I would consider many of the bolded and boxed claims to be too strong.

I do not think that the argument about reasoning traces is strong. My understanding is that the authors took the chain-of-thought from DeepSeek-R1, tokenized it with the LLMTime scheme, and then measured its stationarity. I agree that this is potentially an interesting as a way to measure derailment during extended conversations, but the methods and source of the claims was not clear.

**Questions:**

Have you checked whether nonstationarity depends on the sampling procedure? I would expect the issue to worsen as the sampling temperature increases for probabilistic models. Similarly, how much does the apparent nonstationarity depend on the tokenization scheme?

Since the context is finite length, we can estimate certain quantities like the mean and standard deviation only to limited accuracy. As a result, I wouldn’t necessarily be that surprised if the mean (drift) is slightly off from zero, to within the error that can be narrowed from the context. Does AD Fuller alone account for this?

Can you please provide additional details about the DeepSeek experiments? I understand the template argument broadly, but I don't see how the highlighted reasoning trace supports this description.

---

### Official Review · Reviewer_NtK8 · 2025-10-25

**Soundness:** 3
**Presentation:** 3
**Contribution:** 3
**Rating:** 4
**Confidence:** 5

**Summary:**

This paper examines whether large language models (LLMs) can serve as effective time series forecasters through the lens of stationarity.
By categorizing datasets into stationary and non-stationary groups and systematically evaluating GPT-family and LLaMA models under zero/few-shot settings, the authors demonstrate that LLMs perform competitively only on stationary time series.
The study further reveals that numerical scaling and data formatting strongly affect model stability.
While LLMs lack robustness against non-stationary data, they exhibit reasoning potential when integrated with statistical preprocessing.
The work emphasizes the need for hybrid approaches that incorporate explicit stationarity awareness into LLM-based TSF.

**Strengths:**

1. The “lens of stationarity” provides a simple yet powerful diagnostic view of LLMs’ forecasting limitations.
2. The Evaluation includes multiple LLMs, TS models, and stationarity-controlled experiments.
3. In evaluation, a clear grouping and normalization analyses are applied to help isolate key influencing factors.

**Weaknesses:**

1.This work focuses solely on purely numeric evaluations, lacking more representative settings for LLM-based TSF, such as context-aided time series forecasting or multimodal TSF.

2.The paper only analyzes the reasoning process of DeepSeek-R1; adding similar analyses for other LLMs would provide a deeper understanding of why LLMs fail.

3.The evaluation lacks motivation. Few studies have explored zero-shot time series forecasting with LLMs—Gruver et al.[1] merely demonstrated LLM`s limited TSF ability, such as capturing simple monotonic patterns. Subsequent methods that preserve the autoregressive nature of LLMs on TSF have mainly focused on fine-tuning them [2] [3] [4]. Analyzing the shortcomings of these approaches would make the paper more solid.

[1] Large Language Models Are Zero-Shot Time Series Forecasters

[2] From News to Forecast: Integrating Event Analysis in LLM-Based Time Series Forecasting with Reflection

[3] Time Series Forecasting as Reasoning: A Slow-Thinking Approach with Reinforced LLMs

[4] AutoTimes: Autoregressive Time Series Forecasters via Large Language Models

**Questions:**

Have the authors considered to analyze why LLMs fail in zero-shot TSF from the perspective of the pre-training with “next-token prediction” paradigm? For example, a small numerical change like “1.32” → “1.29” has negligible impact on a time series scaled in [-10, 10], but from a tokenization perspective (where numbers are split into digit tokens), it represents a 50% change. Could this be a primary reason for LLMs’ difficulty in generating and understanding purely numeric sequences?

---

### Official Review · Reviewer_WsWk · 2025-10-26

**Soundness:** 2
**Presentation:** 2
**Contribution:** 2
**Rating:** 2
**Confidence:** 4

**Summary:**

1. Introduces stationarity preservation (mean, variance, autocovariance) as a robustness criterion for zero-shot forecasting

2. Demonstrates that LLMs (e.g., GPT-4, Llama-2) fail to preserve stationarity, with forecasts contaminated by hidden biases and oversimplified reasoning

3. Uses hypothesis testing  on synthetic and real-world datasets

**Strengths:**

1. First to formalize stationarity preservation as a reliability metric for zero-shot LLM forecasting

2. Identifies hidden biases and reasoning flaws

3. Rigorous experiments across 9 LLMs and 5 baselines, with ablation studies on synthetic/real data

4. Well-structured with clear visualizations and pseudocode

**Weaknesses:**

1. The paper uses STL decomposition for real data, but does not explore how LLMs perform on raw non-stationary series, which is common in practice.

2. Focuses exclusively on weak stationarity (mean/variance invariance). While justified, it ignores other properties (e.g., higher-order moments, nonlinear dependencies) that might better capture LLM limitations.

3. Baselines (ARIMA, SMA, etc.) are overly simplistic. Comparisons to more zero-shot methods would strengthen the critique.

4. The observed "hidden trends"  are empirically documented but not theoretically explained

**Questions:**

1. How might LLMs perform on non-stationary series (e.g., stock prices without differencing)?

2. Have the authors explored fine-tuning or prompting strategies to improve LLM performance?

3. Are there other criteria besides stationarity that can further reveal the limitations of LLM?

4. The code will be released, but what were the exact prompts/templates used for LLMs? Small wording changes might significantly affect forecasts.

---

### Official Review · Reviewer_FLWA · 2025-11-01

**Soundness:** 3
**Presentation:** 2
**Contribution:** 2
**Rating:** 6
**Confidence:** 3

**Summary:**

This paper proposes to test the reliability of LLM zero-shot time series forecasters by whether they preserve stationarity in the input. The paper shows most existing LLMs fail at this test, and closer inspection reveals hidden biases in their forecasts such as biasing towards positive values or relying on local information.

**Strengths:**

- The paper uses rigorous statistical tests to validate its hypothesis.
- The proposed criterion of preserving stationarity makes sense as a basic sanity of any forecaster.
- The experiments are thorough.

**Weaknesses:**

- As the paper acknowledges, real time series are rarely stationary, often exhibiting trends or seasonality. This makes the lack of stationarity preservation rarely relevant for practical purposes since stationarity does not hold in the data. Indeed, one of the appeals of LLM forecasters is that they can sometimes zero-shot handle the non-stationarity in the time series without special pre-processing such as the seasonal-trend decomposition.
- The presented argument against LLM forecasters due to lack of stationarity preservation is similar to that made to favor CNNs over Transformers for computer vision due to their baked-in translational equivariance. This lack of translational equivariance did not prevent vision transformers from dominating computer vision today. This reflects a general theme in deep learning where architectures that bake in hard constraints based on initially sensible theoretical considerations are eventually outperformed by those more flexible alternatives capable of handling a wider range of realistic data.
- It is not surprising that baselines such as ARIMA, designed to preserve stationarity, do better than LLMs in this respect. If we know that the time series is (approximately) stationary, we can similarly inform the prediction of LLM forecasters with this knowledge, for example, simply by prompting. This would have been a fairer comparison.

**Questions:**

Why does the reliance on local information make LLM prediction unreliable, especially compared to more traditional methods that are similarly built on locality? For example, ARIMA forecasts using only a small, fixed set of recent values and shocks (AR/MA terms), with effects that decay exponentially as lags get farther away. Even with differencing and seasonal components, it still leans on (seasonally) nearby observations rather than distant history

---

### Meta-Review · Area_Chair_r7Zx · 2026-01-07

**Summary:**

Reviewers agreed that the paper offers a clean empirical framework for testing whether zero-shot LLM forecasters preserve stationarity (mean/variance/autocovariance) using standard hypothesis tests, and that the experiments reveal systematic drift and second-moment failures across multiple LLMs relative to simple statistical baselines. However, the main concerns were about relevance, evaluation setting, and the strength of claims. Several reviewers noted that real-world time series are rarely stationary, so “non-stationarity” may not imply misalignment or practical unreliability, and comparisons would be fairer if LLMs were prompted with stationarity assumptions when applicable. Others questioned the setup (LLMTime + chat models) as atypical for forecasting practice and asked for comparisons against modern zero-shot time-series forecasters (for example, Chronos/Times-FM/Moirai), plus tests on raw non-stationary series rather than STL-detrended data. Finally, reviewers flagged overselling and unclear “reasoning trace” evidence (including sensitivity to sampling temperature/tokenization), limited theoretical explanation for the “hidden trends,” and limited mechanistic analysis beyond DeepSeek-R1.

**Reviewer Scores:**

The vast majority of reviewers gave relatively negative scores. Although some reviewers provided possible reasons for score improvement, it seems that the author team did not properly address all the reviewers' questions during the discussion stage. Therefore, this paper still needs to consider many issues that require improvement.

---

### Decision · Program_Chairs · 2026-01-26

Reject